# Realization of Efficient Phosphorescent Organic Light-Emitting Devices Using Exciplex-Type Co-Host

**DOI:** 10.3390/mi13010051

**Published:** 2021-12-29

**Authors:** Lishuang Wu, Huiwen Xu, Huishan Yang

**Affiliations:** 1Key Laboratory of Information Functional Material for Fujian Higher Education, College of Physics and Information Engineering, Quanzhou Normal University, Quanzhou 362000, China; lishuangw@126.com; 2College of Information Science and Engineering, Huaqiao University, Xiamen 361021, China; xhw15606900927@163.com

**Keywords:** exciplex, reverse intersystem crossing (RISC), thermally activated delayed fluorescence (TADF), efficiency

## Abstract

High-performance phosphorescent organic light-emitting devices with an exciplex-type co-host were fabricated. The co-host is constituted by 1,3,5-tris(N-phenylbenzimidazol-2-yl) benzene, and 4,4,4-tris (N-carbazolyl) triphenylamine, and has obvious virtues in constructing efficient devices because of the thermally activated delayed fluorescence (TADF) resulting from a reverse intersystem crossing (RISC) process. The highest external quantum efficiency and luminance are 14.60% and 100,900 cd/m^2^ for the optimal co-host device. For comparison, 9.22% and 25,450 cd/m^2^ are obtained for a device employing 4,4,4-tris (N-carbazolyl) triphenylamine as a single-host. Moreover, the efficiency roll-off is notably alleviated for the co-host device, indicated by much higher critical current density of 327.8 mA/cm^2^, compared to 120.8 mA/cm^2^ for the single-host device. The alleviation of excitons quenching resulting from the captured holes and electrons, together with highly sufficient energy transfer between the co-host and phosphorescent dopant account for the obvious boost in device performances.

## 1. Introduction

Phosphorescent organic light-emitting devices (PHOLEDs) have tremendous potential in solid-state lighting and flat-panel displays due to the exciton utilization efficiency of 100% in theory realized by harvesting both the singlet and singlet excitons for electro-luminescence devices [1]. Generally, a host–guest system is indispensable for a high-performance phosphorescent device to avoid the concentration quenching, and triplet-triplet annihilation (TTA), and to prompt the device efficiency and stability [2,3].

Host materials are required to possess matchable highest occupied molecular orbital (HOMO), and lowest unoccupied molecular orbital (LUMO) levels to facilitate the charge injection, and sufficiently high triplet energy to ensure the efficient energy transfer from hosts to phosphorescent guests, and effectively confine the excitons [4,5]. More crucially, efficient hosts must have bipolar charge-transporting characteristics to improve carrier balance in emitting layers, and broaden recombination zones [6,7]. Compared to their unipolar counterparts that can efficiently transport only holes or electrons, bipolar host materials, generally having both donor and acceptor moieties in one molecule, are capable of improving the carrier balance in emitting layers, and broadening the recombination zones, promoting device efficiency while suppressing efficiency roll-off at a high operating current density [8,9]. Additionally, using bipolar host materials can simplify the structure of phosphorescent OLEDs [10]. However, developing efficient bipolar host materials to realize an efficient phosphorescent OLED with low operating voltage, high power efficiency, and slight efficiency roll-off is still highly challenging.

Recently, exciplex-type co-hosts composed of acceptors and donors have been employed in highly efficient phosphorescent OLEDs, derived from their unique advantages of improved charge balance and exciton utilization [11]. Significantly, an exciplex system generally has a thermally activated delayed fluorescence (TADF) effect originating from the reverse intersystem crossing (RISC), due to the small singlet-triplet difference (△Est) [12]. Triplet excitons are capable of upconverting into singlet excitons as a result of the intrinsically small △Est. Therefore, the Förster energy transfer will be considerably prompted, boosting the device efficiency, and suppressing the efficiency roll-off. In the present study, we introduced an exciplex-type co-host consisting of a donor of 4,4,4-tris (N-carbazolyl) triphenylamine (TCTA) and an acceptor of 1,3,5-tris (N-phenylbenzimidazol-2-yl) benzene (TPBI) to fabricate efficient yellow phosphorescent OLEDs. The TADF effect was solidly confirmed using the time-resolved photoluminescence spectra technology. Employing this exciplex-type co-host, we realized a high-performance yellow phosphorescent OLED with a maximum external quantum efficiency of 14.60%, and a power efficiency of 45.9 lm/W. In contrast, the peak external quantum efficiency and power efficiency are 9.22% and 27.5 lm/W for the single-host device. Notably, the maximum luminance of the co-host device reaches 100,900 cd/m^2^, about four times that of the single device.

## 2. Experimental Details

The OLEDs were fabricated on the pre-patterned ITO coated glasses. Prior to being deposited into the deposition chamber, the substrates were cleaned with detergent solution (Decon 90), de-ionized (DI) water, isopropyl alcohol, and ethanol, using the heated ultra-sonication baths for 15 min. Then the substrates were blown dry with pure nitrogen gas, and dried in an oven at 120 °C for 30 min, followed by exposure to an oxygen plasma atmosphere for 10 min. The doping process is realized by co-evaporation technology, in which the doping concentration is determined by the deposition rate. All thermal deposition was conducted under a high vacuum of about 3 × 10^−5^ Pa.

The photoluminescence (PL) spectra measurements were undertaken on an Edinburgh Instruments FLS920 spectrophotometer (E.I. Ltd., Edinburgh, UK). The absorption spectra were obtained using a PerkinElmer Lambda750 UV/VIS/NIR spectrometer (P.E. Ltd., Waltham, MA, USA). The phosphorescence spectrum was measured at a low temperature of 77 K after a delay of 100 ms, utilizing the time-resolved emission spectrum (TRES) mode on an Edinburgh FLS980 Spectrometer. The time-resolved photoluminescence spectra were obtained adopting the time-correlated single-photon counting method on an Edinburgh FLS980 Spectrometer. Current density(J)-voltage(V)-luminescence(L) characteristics of all devices were collected simultaneously using a computer-controlled system consisting of a Keithley2400 Source Meter and Minolta Luminance Meter LS-110, and the EL spectra were measured by a PR655 SpectroScan spectrophotometer.

## 3. Results and Discussion

All organic materials were purchased from Shenzhen PURI Materials Technologies Co., Ltd. (Shenzhen, China), of which the molecular structures are presented in Figure 1a,b. Figure 1c presents the PL spectra of TCTA, TPBI, and the mixed film of TCTA and TPBI with the ratio of 1:1. The PL spectrum of TCTA: TPBI (1:1), considerably wider compared to that of TCTA and TPBI, peaks at 440 nm, and is significantly red-shifted relative to 391 nm for TCTA, and 380 nm for TPBI. Moreover, the optical bandgap of TCTA: TPBI (1:1), calculated as 2.85 eV according to the PL peak, is nearly the same as the difference between the HOMO level of TCTA (−5.7 eV) and the (LUMO) level of TPBI (−2.7 eV). As a result, the emission of 440 nm is derived from the new generating exciplex existing between TCTA and TPBI after excitation [13]. Figure 1d depicts the absorption spectra of TCTA, TPBI, and their mixed film. It is observed that there are no additional features in the absorption spectrum for the mixed film in contrast to the pure layers of TCTA and TPBI, meaning no ground state charge transfer (CT) complex exists in the mixed film [14]. Figure 2a,b demonstrate the time-resolved photoluminescence spectrum of TCTA:TPBI (1:1). A prompt and delayed lifetimes of 42.7 ns and 1.16 ms are obtained, respectively, via adopting the double exponential decay model of A + B_1_exp(−t/τ_1_) + B_2_exp(−t/τ_2_), attributed to the prompt fluorescence and delayed fluorescence of the TCTA: TPBI exciplex [15].

The exciplex-based devices using TCTA: TPBI as a co-host were built with the structures of ITO/HAT-CN (5 nm)/TAPC (60 nm)/TCTA (5 nm)/PO-01: TCTA:TPBI (x%, 1:1, 20 nm)/TPBI (60 nm)/LiF (1 nm)/Al. X stands for 3, 6, 9, and 12. For comparison, the reference devices employing the single host of TCTA were also constructed, with the structures of ITO/HAT-CN (5 nm)/TAPC (60 nm)/TCTA (5 nm)/PO-01: TCTA (x%, 20 nm)/TPBI (60 nm)/LiF (1 nm)/Al. HAT-CN and TAPC, standing for 1, 4, 5, 8, 9, 11-Hexaazatriphenylene-hexacarbonitrile and 1,1-bis-(4-bis(4-methyl-phenyl)-amino-phenyl)-Cyclohexane, serve as the hole-injection layer (HIL) and hole-transporting layer (HTL), respectively, whereas TCTA and TPBI function as the hole-transporting type host and electron-transporting layer (ETL), standing for 4,4,4-tris (N-carbazolyl) triphenylamine and 1,3,5-tris(N-phenylbenzimidazol-2-yl) benzene. PO-01, referring to iridium (III) bis(4-phenylthieno[3 ,2-*c*] pyridinato-N, C20) acetylacetonate, is a commonly-used yellow phosphorescent emitter. Figure 3a,b illustrate the current density-voltage-luminescence properties of the fabricated devices. Distinctly shown in Figure 3a, the co-host devices have significantly higher operating current density at the same voltages relative to the reference devices. The boost in current density is attributed to the improved electron mobility resulting from the introduction of TPBI, as well as the alleviation of the charge-trapping effect in the co-host devices [16,17]. Similar promotion in luminance for the co-host devices is also clearly displayed in Figure 3b. Significantly, the maximum luminescence of the co-host devices is 80,800 cd/m^2^, 100,900 cd/m^2^, 104,100 cd/m^2^, and 115,900 cd/m^2^, respectively, notably higher than 18,620 cd/m^2^, 25,450 cd/m^2^, 36,970 cd/m^2^, and 41,760 cd/m^2^ for the single-host devices. The charge-trapping effect in the co-host devices is notably weaker than that in the co-host devices, resulting in the much lower in-built electric field intensity within the emitting zone. Consequently, a considerably higher driving voltage and luminance are achieved for the co-host devices.

Figure 3c,d show the current efficiency-current density-power efficiency curves of the co-host and reference devices. The co-host devices possess much superior current efficiency than the reference devices. For example, the peak current efficiencies for the co-host devices are 44.66 cd/A, 44.69 cd/A, 42.91 cd/A, and 38.93 cd/A, respectively, compared to 22.12 cd/A, 28.23 cd/A, 20.82 cd/A, and 21.35 cd/A obtained for the reference devices. Besides the maximum current efficiency, the current efficiencies under the given high current densities for the co-host devices are also notably better than those for the single-host devices, illustrated in Table 1 summarizing the key EL performances of the co-host and reference devices. Similarly, the improvement in power efficiency can also be observed for the co-host devices. The exciton utilization in the single-host devices is damaged because of the non-radiative T1 of TCTA [18]. In contrast, the non-radiative T1 is capable of upconverting to the radiative efficiently S1 due to the RISC process in the co-host devices, resulting in much higher exciton utilization, since the singlet excitons formed via RISC give their energy to PO-01 molecules by Förster energy transfer [19,20].

Thus, the boost in exciton utilization for the co-host devices accounts for the notable improvement in efficiencies. Figure 4a–d depicts the external quantum efficiency (EQE)-current density (J) plots of the co-host and reference devices. An EQE is calculated according to the assumption of Lambertian distribution, and can be calculated according to the formula [21]:(1)$$\mathrm{EQE}=\frac{\pi \cdot L\cdot e}{683\cdot I\cdot h\cdot c}\;\frac{\int_{380}^{780}I\left(\lambda \right)\cdot \lambda d\lambda }{\int_{380}^{780}I\left(\lambda \right)\cdot K\left(\lambda \right)d\lambda }$$
where *L*, *I*, *λ*, *I*(*λ*) stand for luminance, driving current, wavelength, and the EL intensity, respectively; *K*(*λ*), *e*, *h*, and *c* represent the Commision International de L’Eclairage chromaticity (CIE) standard photopic efficiency function, electron charge, Planck’s constant, and light velocity. Significantly, the co-host devices have better properties in efficiency roll-off at a high current density. For instance, the critical current density (J_0_) for the co-host device with the ration concentration of 6% is as high as 355.4 mA/cm^2^, about 2.9-folds higher than the 120.9 mA/cm^2^ achieved for the reference device with the same doping concentration, meaning the impressive suppression in efficiency roll-off for the co-host device, where J_0_, characterizing the efficiency roll-off property of a phosphorescent OLED is the corresponding current density as EQE declines to half of its initial value obtained without exciton quenching. The exciton recombination zones of the co-host devices are much wider than those of the reference devices, derived from the little charge capture effects [22]. Additionally, the exciton quenching in the co-host devices induced by the excess captured holes at a high current density is considerably alleviated. Consequently, the wide exciton recombination zones, avoidance of captured-holes-induced exciton quenching, and good energy transfer at a high current density are the main factors accounting for the slower efficiency roll-off for the co-host devices [23,24].

## 4. Conclusions

In this study, we solidly confirmed the advantage of the exciplex-type co-host over the single-host in realizing high-performance phosphorescent OLEDs. The co-host, consisting of TCTA: TPBI, forms an exciplex, generating TADF emission derived from the RISC process. The co-host device, with a 6% doping concentration, presents a maximum current efficiency, power efficiency, and external quantum efficiency of 44.7 cd/A, 45.9 lm/W, and 14.6%, respectively. For comparison, the efficiencies for the reference device with single-hosts are 28.2 cd/A, 27.5 lm/W, and 9.2%. Remarkably, the efficiency roll-off at a high current density is considerably suppressed for the co-host devices. The notable improvement in device performance originates from the adequate energy transfer between the exciplex-type co-host and phosphorescent emitter, together with the avoidance of exciton quenching induced by the trapped charges.

## Figures and Tables

**Figure 1 micromachines-13-00051-f001:**
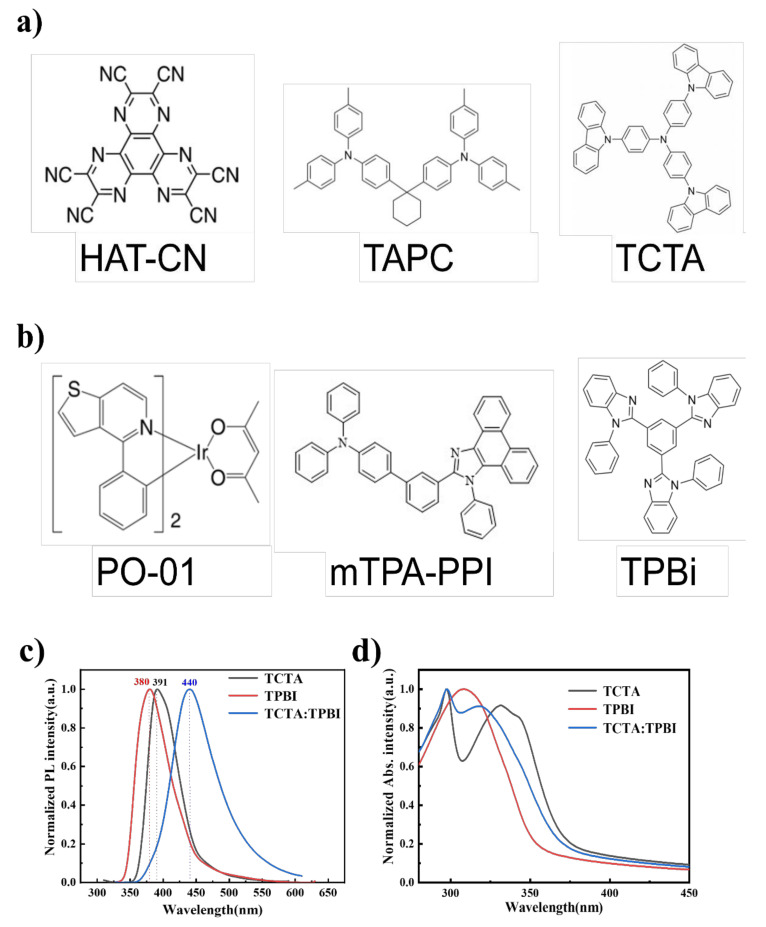
Molecular structures of organic materials used in this study (**a**,**b**), photoluminescence (PL) spectra (**c**), and Abs spectra (**d**) of 4,4,4-tris (N-carbazolyl) triphenylamine (TCTA), 1,3,5-tris (N-phenylbenzimidazol-2-yl) benzene (TPBI), and TCTA:TPBI (1:1).

**Figure 2 micromachines-13-00051-f002:**
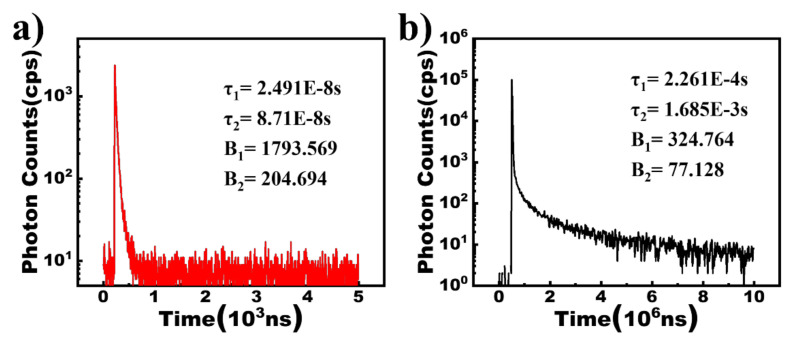
The time-resolved photoluminescence spectra of TCTA: TPBI ((**a**), a short scale time for prompt fluorescence; (**b**), a long scale time for delayed fluorescence); the average lifetimes denoted as τ can be calculated by: $\tau =\frac{\sum B_{i\;}\tau_{i}^{2}}{\sum B_{i\;}\tau_{i}}$.

**Figure 3 micromachines-13-00051-f003:**
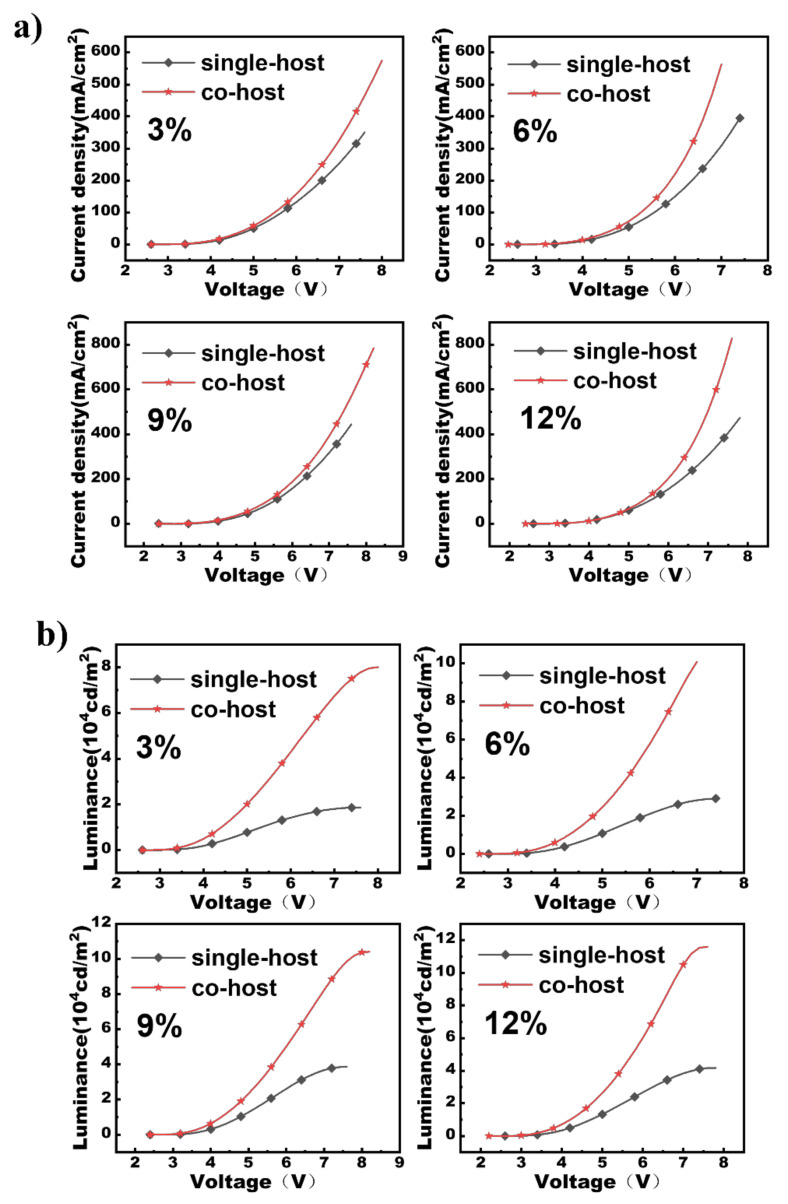
The current density-voltage (**a**), luminance-voltage (**b**), luminance-current efficiency (**c**), luminance-power efficiency (**d**) for the co-host and single-host devices.

**Figure 4 micromachines-13-00051-f004:**
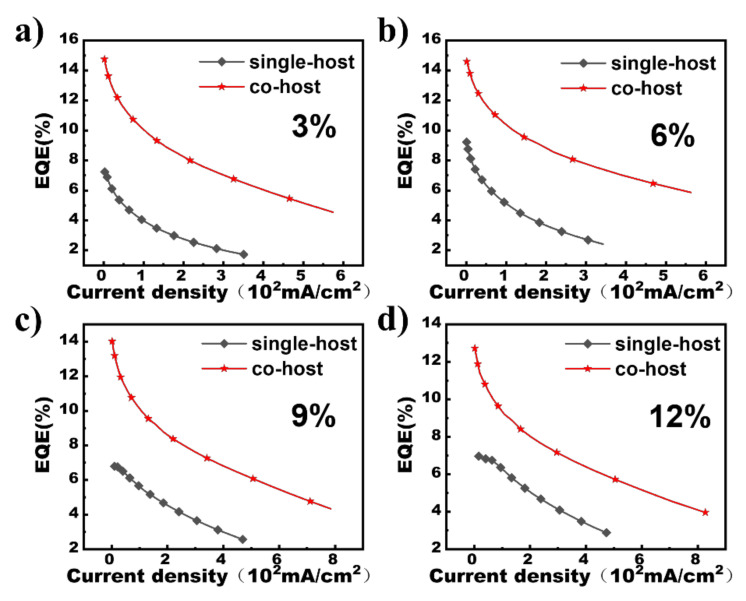
The EQEs of the co-host and single-host devices (**a**–**d**) for 3%,6%,9% and 12% doping concentration, respectively)).

**Table 1 micromachines-13-00051-t001:** The key electroluminescent (EL)performances of the co-host and reference devices.

Device	CE/(cd A^−1^)				PE/(lm W^−1^)				L_max_/(cd m^−2^)		EQE_max_/(%)	
	η_10_^3^		η_max_		η_10_^3^		η_max_					
	Single-Host	Co-Host	Single-Host	Co-Host	Single-Host	Co-Host	Single-Host	Co-Host	Single-Host	Co-Host	Single-Host	Co-Host
3%	22.00	44.46	22.12	44.66	18.40	40.95	20.09	43.77	18,620	80,080	7.23	14.75
6%	27.05	44.48	28.23	44.69	22.72	42.10	27.52	45.87	25,450	100,900	9.22	14.60
9%	20.62	42.75	20.82	42.91	17.43	41.12	17.97	43.53	36,970	104,100	6.80	14.02
12%	18.09	38.15	21.35	38.93	16.20	37.15	17.61	37.32	41,760	115,900	6.97	12.72

## Data Availability

Not applicable.

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
