# Peer review of "Realization of Efficient Phosphorescent Organic Light-Emitting Devices Using Exciplex-Type Co-Host"

_micromachines, 2021, doi:10.3390/mi13010051_

Round 1

Reviewer 1 Report

The study involves investigation of a co-host formulation to act as a OLED. It is clear from the results shown that the co-host exciplex represents an improvement in efficiency and luminance, versus the single excimer materials. The study is sound and the conclusions drawn are in-line with the results presented.

Needs attention:

Experimental Details should include a brief description of the 3, 6, 9 and 12% doping concentrations, as these are shown in figures and referred to sporadically in text, but not described.

Minor:

line 76 - missing parenthesis.

Fig 1 quotes 440nm peak for TCTA:TPBI - which looks correct, while text refers to 436nm.

line 147 - 'Ldamaged' typo.

line 152 - table description error

line 164 - "For an instance" should be "for instance"

line 182 - should begin with "In this study"

Author Response

Dear Reviewer,

We want to express our appreciation for your constructive comments regarding our manuscript entitled “Realization efficient phosphorescent organic light-emitting devices using exciplex-type co-host” (Manuscript ID: 1484784)”. The comments are highly beneficial for us to revise and improve the paper. We have studied the suggestion carefully and have changed the manuscript according to the comments. The modified portions are marked in red in the revised manuscript.

(Q: The comments; R: Our response)

To Reviewer #1:

Q: The study involves investigation of a co-host formulation to act as a OLED. It is clear from the results shown that the co-host exciplex represents an improvement in efficiency and luminance, versus the single excimer materials. The study is sound and the conclusions drawn are in-line with the results presented.

Needs attention:

Experimental Details should include a brief description of the 3, 6, 9 and 12% doping concentrations, as these are shown in figures and referred to sporadically in text, but not described.

Minor:

line 76 - missing parenthesis.

Fig 1 quotes 440nm peak for TCTA: TPBI - which looks correct, while text refers to 436nm.

line 147 - 'Ldamaged' typo.

line 152 - table description error

line 164 - "For an instance" should be "for instance"

line 182 - should begin with "In this study"

  1. Special thanks for your constructive suggestion. According to your suggestion, we have illustrated the brief description of different doping concentration, as marked in red in line 69. Moreover, we have checked the manuscript carefully and revised all minor.

We are deeply grateful to the reviewer for giving us the opportunity to revise the submitted paper. We tried our best to improve the manuscript and made some changes in the manuscript, and hope the revision will meet with approval. Thank you very much for your comments and suggestions again.

Wishing you good health during this difficult time.

Yours sincerely,

 Prof. Huishan Yang

Reviewer 2 Report

I have no comments at the moment.

Author Response

We want to express our appreciation for your constructive comments regarding our manuscript entitled “Realization efficient phosphorescent organic light-emitting devices using exciplex-type co-host” (Manuscript ID: 1484784)”. The comments are highly beneficial for us to revise and improve the paper. We have studied the suggestion carefully and have changed the manuscript according to the comments. The modified portions are marked in red in the revised manuscript.

(Q: The comments; R: Our response)

To Reviewer #2: During our initial check, we notice that some sentences in your manuscript are similar with the contents in your already published paper (see the

highlight of the attachment), please rephrase and check your whole manuscript

during the revision.

  1. According to your suggestion, we have changed the similar contents.

We are deeply grateful to the reviewers for giving us the opportunity to revise the submitted paper. We tried our best to improve the manuscript and made some changes in the manuscript, and hope the revision will meet with approval. Thank you very much for your comments and suggestions again.

Wishing you good health during this difficult time.

Yours sincerely,

 Prof. Huishan Yang

Reviewer 3 Report

The manuscript titled “Highly efficient phosphorescent organic light-emitting device using exciplex-type co-host (Lishuang Wu, Huiwen Xu, and Huishan Yang)” is very interesting and attractive to many researchers. The manuscript is clear and well structured. The research problem is well explained and supported by appropriate experiments. I recommend this manuscript for publication.

Author Response

We want to express our appreciation for your constructive comments regarding our manuscript entitled “Realization efficient phosphorescent organic light-emitting devices using exciplex-type co-host” (Manuscript ID: 1484784)”. The comments are highly beneficial for us to revise and improve the paper. We have studied the suggestion carefully and have changed the manuscript according to the comments. The modified portions are marked in red in the revised manuscript.

(Q: The comments; R: Our response)

To Reviewer #3:

Q: The manuscript titled “Highly efficient phosphorescent organic light-emitting device using exciplex-type co-host (Lishuang Wu, Huiwen Xu, and Huishan Yang)” is very interesting and attractive to many researchers. The manuscript is clear and well structured. The research problem is well explained and supported by appropriate experiments. I recommend this manuscript for publication.

  1. The authors are greatly grateful for the constructive suggestions from reviewers.

We are deeply grateful to the reviewers for giving us the opportunity to revise the submitted paper. We tried our best to improve the manuscript and made some changes in the manuscript, and hope the revision will meet with approval. Thank you very much for your comments and suggestions again.

Wishing you good health during this difficult time.

Yours sincerely,

 Prof. Huishan Yang

Reviewer 4 Report

TCTA, TPBI are used to form OLED devices, but there is no information about the material. Why did you choose these materials and where did you purchase them? 

It would be easier to understand the structures of the devices if you drew the materials used in OLED devices.

Author Response

We want to express our appreciation for your constructive comments regarding our manuscript entitled “Realization efficient phosphorescent organic light-emitting devices using exciplex-type co-host” (Manuscript ID: 1484784)”. The comments are highly beneficial for us to revise and improve the paper. We have studied the suggestion carefully and have changed the manuscript according to the comments. The modified portions are marked in red in the revised manuscript.

(Q: The comments; R: Our response)

To Reviewer #4:

Q: TCTA, TPBI are used to form OLED devices, but there is no information about the material. Why did you choose these materials and where did you purchase them? 

It would be easier to understand the structures of the devices if you drew the materials used in OLED devices.

  1. The authors agree with the scientific suggestion proposed by the reviewer. All organic material were purchased from Shenzhen PURI Materials Technologies Co., Ltd and were used without purified. Additionally, their molecular structures were depicted in Figs.1a-1b.

We are deeply grateful to the reviewers for giving us the opportunity to revise the submitted paper. We tried our best to improve the manuscript and made some changes in the manuscript, and hope the revision will meet with approval. Thank you very much for your comments and suggestions again.

Wishing you good health during this difficult time.

Yours sincerely,

 Prof. Huishan Yang

Round 2

Reviewer 2 Report

No one.